

# Root distributions predict shrub-steppe responses to precipitation intensity

Andrew Kulmatiski[1], Martin C. Holdrege[1], Cristina Chirvasa[1], Karen H. Beard[1]

[1]Department of Wildland Resources, Utah State University and the Ecology Center, Logan, UT 84322-5230 USA

*Correspondence to:* Andrew Kulmatiski (andrew.kulmatiski@usu.edu)

**Abstract.** Precipitation events are becoming more intense around the world, changing the way water moves through soils

10   and plants. Plants that have, or create, roots that absorb more water under these conditions are likely to become more abundant (e.g., shrub encroachment). Yet, it remains difficult to predict species responses to climate change because we typically do not know where active roots are located or how much water they absorb. Here, we used water tracer injections in a field experiment to describe forb, grass, and shrub root distributions under low and high precipitation intensity treatments. To estimate how much water different active rooting distributions can absorb over time, we used a soil water

15   flow model, and we compared our estimates of water uptake to aboveground plant growth. In low precipitation intensity plots, deep shrub roots were estimated to absorb the most water (93 mm yr$^{-1}$) and shrubs had the greatest aboveground cover (27%). Grass root distributions were estimated to absorb an intermediate amount of water (80 mm yr$^{-1}$) and grasses had intermediate aboveground cover (18%). Forb root distributions were estimated to absorb the least water (79 mm yr$^{-1}$) and had the least aboveground cover (12% cover). In high precipitation intensity plots, shrub and forb roots moved in ways that

20   increased their water uptake relative to grasses, predicting the increased aboveground growth of shrubs and forbs in these plots. In short, water uptake caused by different rooting distributions predicted plant aboveground cover. Our results suggest that detailed descriptions of active plant root distributions can predict plant growth responses to climate change in arid and semi-arid ecosystems.

## 1 Introduction

Changes in precipitation patterns are an important component of climate change (Humphrey et al., 2021; Jiao et al., 2021; Liu et al., 2020b). Precipitation intensification, in particular, has been predicted and observed around the world (Liu et al., 2020a; Giorgi et al., 2011). Precipitation intensification is predicted to continue because warmer air can hold more moisture (i.e., roughly 7% per °C) resulting in fewer, but larger precipitation events (O'Gorman and Muller, 2010; Trenberth et al., 2003). The implications of precipitation intensification for vegetation remain difficult to resolve with some species and ecosystems responding positively and others responding negatively (Good and Caylor, 2011; Knapp et al., 2008; Liu et al., 2018).

In arid and semi-arid areas, small (~1-5 mm) events are common and often evaporate before reaching the rooting zone (Lauenroth and Bradford, 2009). With larger precipitation events (~5-20 mm), less water is lost to evaporation and water percolates deeper into the soil (Sala et al., 2015). In arid systems, decreased evaporation is likely to increase growth of shallow-rooted plants (e.g., grasses; Post and Knapp, 2021). In semi-arid systems, deeper percolation is likely to increase growth of deeper-rooted species (e.g., shrubs; Gherardi and Sala, 2015; Xu, Medvigy and Rodriguez-Iturbe, 2015; Holdrege, Beard and Kulmatiski, 2021). In mesic systems, however, increasing event size may increase runoff and percolation below the rooting zone. This may have little effect on plant productivity or lead to tree encroachment (Jones et al., 2016; Slette et al., 2022; Berry and Kulmatiski, 2017; Knapp et al., 2008).

Thus, predicting whether different plants will grow more or less in response to precipitation intensification, requires an understanding of both soil water flow and rooting distributions (D'Onofrio et al., 2019; Schreiner-McGraw et al., 2020; Slette et al., 2022; Xu et al., 2015). Soil water flow models have become very effective at describing water movement through the soil profile and into plant roots (Šimůnek et al., 2016). It is fairly easy to parameterize and validate these models in plant monocultures or at the ecosystem level. It is much more difficult to understand how different species will respond to climate change because it remains difficult to distinguish roots of different plant species and the amount of water absorbed by the roots of different species (Case et al., 2020; Smithwick et al., 2014; Erktan et al., 2018).

Root biomass distributions have long been used to infer the timing and extent of water uptake (Silvertown et al., 2015).

However, there are many reasons that root biomass may not be correlated with water uptake (Cai et al., 2018). Many roots

are structural and do not actively absorb water (Zarebanadkouki et al., 2019). Even fine roots that do absorb water, cannot

absorb water from dry soils (Šimůnek et al., 2016). Further, root water potential, aquaporin density or activity, and stem

conductance can all affect water uptake rates (Dybzinski et al., 2019; Zarebanadkouki et al., 2019).

Natural abundance stable isotope techniques provide a clearer picture of water uptake than root biomass, but

typically only distinguish 'shallow' from 'deep' root distributions (Dubbert and Werner, 2019; Rothfuss and Javaux, 2017).

This is a problem because small differences in root distributions can have large effects on water uptake (Kulmatiski et al.,

2020b). Depth-controlled tracer techniques can provide more detailed descriptions of active root distributions, but are more

difficult to perform, resulting in smaller sample sizes (Rothfuss and Javaux, 2017; Rasmussen and Kulmatiski, 2021; Case et

al., 2020; Holdo and Nippert, 2022). Neither of these techniques, however, estimate the total amount of water moving

through plant roots (McMurtrie and Näsholm, 2018; Rasmussen and Kulmatiski, 2021). Estimates of total water uptake

require other techniques, such as stemflow measurements or the use of soil water flow models to estimate water uptake over

time (Cai et al., 2018; Schymanski et al., 2008). Yet, few studies measure active rooting distributions, let alone the amount

of water absorbed by these distributions (Dybzinski et al., 2019). One recent study that did estimate water uptake found that,

contrary to long held assumptions, deep roots typically absorb more water than shallow roots (Kulmatiski and Beard, 2022).

One important additional complexity in understanding root water uptake is that root distributions change over time

(Barberon et al., 2016; Guderle et al., 2018; Schymanski et al., 2008). Thus, it is important to measure root activity over time

and as resource availability changes (J. Liu *et al.*, 2020). Plants that can adapt to changes in resource availability are more

likely to have positive growth responses than plants with fixed rooting patterns (Berry and Kulmatiski, 2017; Zhou et al.,

2019).

Our goal was to test if root distributions could predict plant growth and responses to precipitation intensity. To do

this, 1) we measured tracer uptake by forbs, grasses and shrubs in the fifth year of a six-year field experiment with low- and

high- precipitation intensity plots, 2) we used a soil water flow model to estimate water uptake by these different root

distributions under low- and high- precipitation intensity conditions, and 3) we compared our estimates of water uptake to

plant growth in the field experiment. We performed this research in an experimental setting where a previous paper reported

that three years of treatment with larger precipitation events 'pushed' water deeper into the soil and increased shrub growth

(Holdrege et al., 2021). Here, we report vegetation responses for an additional two treatment years and one post-treatment

year. Broadly, this approach allowed us to describe the hydrological niches of forbs, grasses and shrubs. It also allowed us to

test if the amount of water these niches could absorb predicted plant growth observed in a field experiment.

**1.       Study area and methods**

1.1     *Study area*

We conducted this research at the Hardware Ranch Wildlife Management Area (41° 36' 53" N, 111° 34' 1" W; 1760 m),

Utah, USA. Mean annual precipitation is 468 mm, with 170 mm (36%) falling as snow, primarily between December and

March (Global Historical Climatology Network - Daily (GHCN-Daily), Version 3). Mean monthly temperatures range from

-4 °C in January to 23 °C in July (Global Historical Climatology Network - Daily (GHCN-Daily), Version 3). Soils are

derived from quartzite and sandstone and are in the Yeates Hollow series (well-drained, cobbly silty clay loam; Soil Survey

Staff, 2018).

Common plant species in this sagebrush-dominated rangeland include shrubs: big sagebrush (*Artemisia tridentata*

Nutt. ssp. *vaseyana* [Rydb.] Beetle; 25% cover), bitterbrush (*Purshia tridentata* [Pursh] DC.; 4% cover), rabbitbrush

(*Chrysothamnus viscidiflorus* [Hook.] Nutt.; 2% cover), forbs: Aster (*Aster kingii*; 3% cover), western yarrow (*Achillea*

*millefolium*; 3% cover), violet (*viola species*; 2% cover), grasses: meadow brome (*Bromus commutatus* Schrad., 10 %

cover), bluebunch wheatgrass (*Pseudoroegneria spicata* [Pursh] Á. Löve; 6% cover), and prairie Junegrass (*Koeleria*

*macrantha* [Ledeb.] Schult.; 1 % cover). Aboveground net primary productivity at the site is approximately 145 g m$^{-2}$ year$^{-1}$

(Holdrege et al. 2020). Cattle were excluded during the experiment, but livestock have grazed the site for over 100 years.

Native ungulates (elk and deer), rabbits, and rodents are common and accessed plots.

*2.2 Experimental treatments*

In June 2015, we established 14 plots (8 m x 8 m each) in a grid with at least 15 m between plots. Three plots were shelter-free controls. We covered the remaining 11 plots with clear Plexiglass® acrylic (6.35 mm thick) roof (Fig. 1). We collected

rainwater from roofs in two holding tanks per shelter. Float switches and water pumps sprayed collected water through six sprinkler heads (1 m height) at a rate that is in the upper quartile of precipitation intensity for the site (26 mm hour$^{-1}$; Holdrege et al. 2020). We manipulated precipitation between January 2016 and September 2020 (five growing seasons), when shelters were removed. Soil and vegetation responses from the first three growing seasons (2016, 2017, 2018) are reported elsewhere (Holdrege et al. 2020). Here we report vegetation responses for two additional treatment seasons (2019

and 2020) and for one post-treatment season (2021). We also conducted a depth-controlled tracer experiment to measure active root distributions in the last treatment season 2020.

We designed treatments to create precipitation event sizes expected with temperature changes from -1 to +10 °C relative to current temperatures. Consistent with the Clausius-Clapeyron relationship, we designed event sizes to increase by 7% per 1 °C of warming (O'Gorman & Muller, 2010; Holdrege et al. 2020). Thus, temperature increases of 1 °C, 2 °C, 3 °C,

5 °C and 10 °C resulted in minimum precipitation event sizes of 2, 3, 4, 8 and 18 mm. To expand our inference, we added a treatment associated with -1 °C temperature change. In this treatment, we manually triggered irrigation multiple times during each growing season to deposit additional 1 mm events (hereafter referred to as the 1 mm treatment).

We based snow addition frequencies on the historical distribution of snow events >4 cm (Global Historical Climatology Network - Daily (GHCN-Daily), Version 3). We estimated that each 1 °C would result in a 7% change in

precipitation event size which meant there would be a median of 14, 13, 11, 10, 8, 7 and 4 snow events per year for the 1, control, 2, 3, 4, 8, and 18 mm treatments. In control plots, we removed snow (>4 cm) from the roofs and immediately shoveled it back onto the plot. For treatments to receive fewer larger snow events, we removed snow off the shelter roofs and allowed it to accumulate on plastic sheeting adjacent to plots before shoveling it back onto the plots. As with rain, all treatments received the same amount of total snow, and only differed in the timing and magnitude of the events. We

established the experiment using a regression design with a range of precipitation intensities, but here we group data into high (18 mm, 8 mm, 4 mm, 4 mm, and 4 mm) and low (1 mm, control, control, control, 2 mm, 3 mm) precipitation intensity treatments.
### 2.3 Abiotic treatment responses

Shelters caused 0.6 °C warming, but decreased wind speed and increased relative humidity, which resulted in little change in evapotranspiration demand inside and outside of plots (Holdrege et al. 2021). During the first three years (2016-2018), high intensity treatments 'pushed' soil water deeper into the soil and increased soil water availability (presumably by decreasing evaporative losses from small precipitation events [Holdrege et al. 2022]). Here we report soil moisture data from 2019 and 2020 (sensors became unreliable or non-functional in 2021). We measured soil moisture using heat dissipation sensors

located at six depths in one treated and one control plot (229L heat dissipation sensors, Campbell Scientific, Logan, UT, USA; Flint et al. 2002). We converted water potential data to volumetric water content using site-specific soil characteristic curves for shallow (0-30 cm) and deep (30-100 cm) soils. Because soil moisture data are from one plot and lack spatial inference, we report values but do not perform statistical tests for treatment differences.

### 2.4. Active root distributions: tracer uptake

We used a depth-controlled hydrologic tracer experiment to describe the vertical distribution of tracer uptake by forbs, grasses, and shrubs in low- and high- intensity precipitation plots (Kulmatiski *et al.*, 2010). We performed tracer injections in four low-precipitation treatment plots (1 mm, control, control, 2 mm) and four high-precipitation plots (4 mm, 4 mm, 4 mm, 18 mm). Four circular subplots (1.5 m diameter) were located near the center of each of the four quadrants in each 8 m by 8

m plot so that the edge of each subplot was separated by 2.7 m. Previous studies indicated that little to no tracer will be absorbed from 2.7 m away during the two-day sampling period (Holdrege et al., 2021; Kulmatiski et al., 2010; Berry and Kulmatiski, 2017). We randomly assigned each subplot to a target depth (10 cm, 20 cm, 45 cm, 75 cm). We drilled 68 pilot holes in a 15 cm by 15 cm grid pattern to the target depth using a 10-mm drill bit and a hammer drill (TE-60, Hilti North America, Texas). In each pilot hole, we injected 1 mL of 70% deuterium oxide ($D_2O$). To rinse the syringe and prevent

contamination in the injection hole, we then injected 2 mL of rinse water. Though injections certainly increased water availability at the point of injection, the 204 mL of tracer and rinse water injected to each 1.8 $m^2$ plot was equivalent to a
trivial 0.1 mm of precipitation event and was expected to act as a tracer and not a resource pulse. We performed these injections early and late in the growing season, May and July 2020.

Two days following injections, we clipped non-transpiring tissues from dominant species in each plot with triple-rinsed clippers and placed them in custom-made 19-mm wide, medium-walled borosilicate tubes, sealed with parafilm and placed on ice. We often composited samples from multiple individuals of the same species into a single sample tube. We collected replicate composite samples from each species when possible. We brought samples to freezer storage within a few hours. We extracted sample water using batch cryogenic distillation. $D_2O$ concentrations were measured using cavity ring-down spectrometry (Picarro L2120-*i*; CA). Recent studies demonstrating fractionation by plant species and tissues and 155 cryogenic distillation in natural abundance isotope studies are not relevant to this experiment because tracer concentration were orders of magnitude greater than potential fractionation by these sources.

### 2.5. Estimated water uptake

We used tracer uptake to describe the distribution of active roots. To estimate water flow into active roots over time, 160 we used the water flow model Hydrus 1D (Šimůnek, van Genuchten and Šejna, 2016; Mazzacavallo and Kulmatiski, 2015). We first parameterized this model with community-level root data reported in Holdrege et al. (2021). We executed the model for low and high precipitation intensity conditions. We used ambient observed precipitation for low precipitation intensity simulations and a tipping-bucket model that 'collected and redeposited' only 8 mm precipitation events to create precipitation for the high precipitation simulation. We selected the 8 mm precipitation event treatment because it represented 165 a mean precipitation event size among the plots where tracers were applied. This reduced the number of model simulations needed, but we expected it to provide reasonable representation of water flow through soils with small or large precipitation events. These simulations estimate total plant community transpiration. We partitioned this community-level water uptake into forb, grass, and shrub components using the proportion of tracer uptake by depth by different plant growth forms (i.e., assuming symmetric root competition; (Cahill, Jr. and Casper, 2000; Kulmatiski and Beard, 2022). To be clear, we did not 170 use the model to simulate water uptake by forb, grass, or shrub canopies. We were specifically interested in isolating the effects of root distributions on water uptake by a theoretical plant canopy.

More specifically, we performed an 'initialization' model simulation run to identify soil hydraulic parameters and to produce estimates of soil moisture over time that could be validated against observed soil moisture. This initialization run used initially observed soil moisture, new root area, soil texture, soil bulk density, and microclimate data (Holdrege et al. 2021). We used the neural network predictions within Hydrus 1D to select hydraulic parameters. Within Hydrus 1D, the Penman-Monteith sub-model estimated evapotranspiration, the van Genuchten-Mualem water flow sub-model simulated unsaturated soil hydraulic properties, and the Feddez sub-model for alfalfa simulated root water uptake. We assumed plant height to be 60 cm and leaf area was calculated by Hydrus 1D from plant height associated with an alfalfa crop. We used a critical stress index of 1.0 because we measured root distributions directly (Kulmatiski 2020). Again, we held plant height and critical stress values constant for forb, grass, and shrub root distributions so we could isolate the effects of root distributions from the effects of other factors, such as aerodynamic resistance, stomatal conductance, leaf area, leaf water potential, etc. A previous study including these effects (Mazzacavallo and Kulmatiski, 2015) found that root distributions were the dominant factor determining water uptake.

Following the initialization run, we conducted a five-year simulation of community-level water uptake using climate data from 2016-2020. We then multiplied these values by the proportion of forb, grass, and shrub tracer uptake to separate total water uptake into forb, grass, and shrub components. For example, if tracer data indicated that 2%, 2%, and 4% of forb, grass, and shrub tracer uptake occurred at 25-26 cm depths, then we estimated that shrubs absorbed 50% [i.e., 4% / (2% + 2% + 4%)] of total water uptake at that depth (Kulmatiski 2020). This approach produces estimates of water uptake that are consistent with ecosystem-level water flow and correlated with plant landscape abundance (Mazzacavallo and Kulmatiski, 2015; Kulmatiski et al., 2020a). The final product of this approach was estimates of the amount of water forb, grass, and shrub roots can be expected to absorb from different soil depths over time. We repeated this process with precipitation patterns that reflected our high-intensity treatment (Holdrege et al., 2021). Essentially, we used a tipping bucket model that collected observed precipitation and deposited it only as 8 mm events. We partitioned these water uptake values into forb, grass and shrub components using the tracer uptake data collected from the high precipitation intensity plots.

The analyses above describe how each plant type's roots would absorb water if each plant had equal transpiration demand (i.e., equal leaf area, stomatal conductance, aerodynamic resistance, etc.). To provide an estimate of water uptake

that reflected actual water uptake on the landscape, we weighted water uptake by each root distribution by observed leaf percent cover and stomatal conductance. To do this, leaf area was multiplied by stomatal conductance values to create a 'canopy demand' value. Each canopy demand value was divided by the mean canopy demand value to create a multiplier

that adjusted each plant type's uptake demand and also maintained the same amount of total water uptake. We assume aerodynamic resistance is similar among these low-statured plants. We also ignore the potential effects of water use efficiency because whole plant water use efficiency were not available and published data are mixed (Hai et al., 2022; Toft et al., 1989; Golluscio and Oesterheld, 2007).

*2.5. Biotic treatment responses*

Every June (peak growing season), we determined percent cover by plant species using visual estimation in nine, permanent 1 m x 1 m subplots in each plot. We measured shrub stem radius on the main stems of the three *A. tridentata* closest to the center of the plot using point dendrometers mounted 10 cm from the ground (spring return linear position sensor BEI 9605, BEI Sensors, Thousand Oaks, CA, USA; Wang and Sammis 2008). To limit damage caused by mounting sensors onto

stems, we only used stems with a radius > 3.5 cm. We recorded stem radial growth hourly to 0.1 mm (CR10X; Campbell Scientific, Logan, Utah, USA).

In June 2016, we measured stomatal conductance for three forbs, six grasses and 18 shrubs in each plot. In July 2017, we measured stomatal conductance on 18 shrubs in each plot. We used a steady state porometer (SC-1, Decagon Devices, WA) to measure stomatal conductance. Stomatal conductance was measured on both sides of grass and shrub

leaves, but stomatal conductance was not detectable on adaxial surfaces of forbs. Grass and shrub stomatal conductances were doubled to account for conductance on both sides of the leaves.

*2.7. Statistical analysis*

We conducted all analyses using R version 3.4.05 (R Core Team, 2018). Broadly, we analyzed repeated measures

data taken in all plots using generalized additive mixed effects models (GAMMs). For example, tracer uptake by depth was analyzed using GAMMs. Analyses were performed separately for early season-low intensity treatments, early-season-high

intensity treatments, late season-low intensity treatments and late season-high intensity treatments. In each case, a null model where a single spline was fit to depth (no treatments distinguished) was compared to a model that grouped vegetation into plant functional types with three levels (forb, grass, shrub; Wood, 2004). We also analyzed shrub stem radius over time using

GAMMs (mgcv package. We fit two GAMMs that contained the fixed effect of date and random effect of plot: 1) a null model where a single spline was fit to date (no treatments distinguished), and 2) a model that grouped treatments into two levels, low intensity (1 mm, control, 2 mm, and 3 mm treatments) and high intensity (4 mm, 8 mm, and 18 mm treatments). Data smoothing to remove spurious values was performed using a moving $10^{th}$ percentile or $90^{th}$ percentile 'window' and a 24-hour wide bin. Vegetation cover determined by visual estimation in nine fixed quadrats in each plot was similarly

analyzed. Top models were those with the lowest Akaike's information criterion (AIC) and we considered models similar if $\Delta AIC < 2$ (Burnham and Anderson, 2002).

We report daily soil water content measurements made in one control and one treated plot, but we did not statistically compare them due to a lack of spatial replication. Similarly, we did not perform statistical analyses on the water uptake estimates because they are the product of a deterministic model.


## 2. RESULTS

### 3.1. Abiotic responses

In both shallow (0-30 cm) and deep (30-100 cm) soils, soil moisture was greater in high-precipitation intensity treatments than low precipitation intensity treatments (Fig. 2).


### 3.2. Tracer and water uptake

The mean depth of tracer uptake often provides a good index of rooting distribution depths (Fig. 3; Kulmatiski, Adler and Foley, 2020). Broadly, this index revealed that 1) shrub roots were deeper than forb or grass roots, 2) roots moved deeper later in the season, and 3) roots moved shallower in response to greater precipitation intensity. More specifically, early in the

season, the mean tracer uptake depth was deeper for shrubs (25 cm) than for forbs (17 cm) or grasses (14 cm). Later in the season, the mean tracer uptake depth was 58 cm, 21 cm, and 21 cm for shrubs, forbs and grasses, respectively. With the

exception of forbs in May, the mean depth of tracer uptake was deeper (4 cm to 15 cm) in low than high precipitation

intensity plots.

There was similar or more support for models that separated tracer uptake by plant type than for models that

grouped uptake profiles (Fig. 3; Table 1). The only exception was early season, high intensity plots, where there was more

support for the model that grouped uptake profiles. In other words, plant functional groups demonstrated different tracer

uptake patterns, except when soil water availability was large.

Tracer uptake indicated the depth of active roots, but water uptake indicated how much water flowed into those

active roots. The mean depth of soil water uptake (estimated by the soil water flow model) was generally deeper than the

mean depth of tracer uptake because shallow soils would dry and provide little water (Fig. 4). Across the year, the mean

depth for forb, grass, and shrub water uptake was 18 cm, 21 cm and 37 cm in low precipitation intensity plots. In contrast to

tracer uptake, the mean depth of water uptake was deeper in high than low precipitation intensity plots (27 cm, 23 cm, and

39 cm for forbs, grasses and shrubs, respectively; Fig. 4). This occurred because high precipitation intensity treatments

decreased evaporation (from 13 to 8 cm per year) and increased percolation (from 28 to 32 cm per year), resulting in

increased deep soil water availability (Fig. 2). We did not perform statistical tests on water uptake because they are the

product of tracer uptake and a deterministic water flow model.

Finally, we weighted water uptake to account for leaf area and stomatal conductance. This provided an estimate of

the amount of water we expected each plant type to absorb on the landscape. Large shrub and grass leaf area and stomatal

conductance increased the estimated water uptake of these plant types (Fig. 5).


### 3.3. Biotic responses

Shrub stem diameter growth showed a response to treatments, not measured in forbs or grasses. Shrub stem diameter was

greater in high intensity than low intensity plots (Fig. 6), and there was more support for the model separating data from high

and low precipitation intensity treatments (AIC = 1209.7) than for a model that grouped all data (AIC = 2291.9). Shrub stem

diameter growth was 30% higher in high intensity than low intensity plots. Although this effect was not detected with visual

estimation of shrub cover (Fig. 7), suggesting that stem diameter was more sensitive to treatments, where there was more





support for the model grouping shrub cover data from high and low intensity plots (AIC = 397.6) than for a model that separated shrub cover data (AIC = 401.4). Across years, shrub cover was 26.9% across all plots.

Forb cover was greater in high than low precipitation treatments (Fig. 7) but there was equal support for models that

grouped or separated forb cover data from high and low intensity plots (AIC = 375.8 and 377.4, respectively). Across years, forb cover was 13.7% in all plots, 11.6% in the low intensity plots and 15.8% in the high intensity plots. There was more support for a model that grouped grass cover data (AIC = 396.5) than for the model that split grass cover data from high and low intensity plots (AIC = 401.3; Fig. 7). Across years, grass cover was 18.4% across all plots.

In the post treatment year (2021), there were no differences in forb, grass or shrub cover between low intensity and

high intensity plots ($P > 0.$). Shrub stem diameter remained greater in high than low precipitation intensity plots during the post-treatment year: there was more support for a model separating treatments than for the model grouping treatments ($\Delta$AIC = 2047).

Due to conductance on both sides of the leaf, shrub (322 +/- 10 mmol m$^{-2}$ sec$^{-1}$) and grass stomatal conductance (310 +/- 31 mmol m$^{-2}$ sec$^{-1}$) was greater than forb (244 +/- 21 mmol m$^{-2}$ sec$^{-1}$) stomatal conductance. There was no difference

in stomatal conductance between high and low precipitation intensity plots for forbs, grasses, or shrub ($T_{1,6}$, $P > 0.05$ for forbs, grasses and shrubs).

Across treatments, the percent cover each plant type each year was correlated with root water uptake estimated by root distributions alone [cover(%) = -24 + 2.5*water uptake (mm); $F_{1,28}$ = 12.42, $P = 0.002$, $R^2 = 0.31$], and with root water uptake estimated from root distributions weighted by leaf area and stomatal conductance [cover(%) = 8.6 + 0.6*water uptake

(mm); $F_{1,28}$ = 24.78, $P < 0.001$, $R^2 = 0.47$].

## 3.  DISCUSSION

Plant rooting distributions helped predict plant growth under low and high precipitation intensity conditions. Aboveground, shrubs were the dominant lifeform (27% cover) with grasses (18% cover) and forbs (12% cover) less abundant. When we

assumed that these three plant types had the same water demand (i.e., leaf area, stomatal conductance, *etc.*), shrub roots were estimated to absorb 93 mm yr$^{-1}$ which was more than grass (80 mm yr$^{-1}$) or forb (79 mm yr$^{-1}$) rooting distributions. When we

weighted this uptake by leaf area and stomatal conductance, shrubs were estimated to absorb 140 mm yr$^{-1}$, which was larger than grasses (77 mm yr$^{-1}$) or forbs (42 mm yr$^{-1}$). Therefore, whether weighted by existing plant cover or not, root distributions predicted that shrubs could absorb the most water followed by grasses and forbs, which correctly predicted the

rank order abundance of aboveground plant cover. Shrub roots absorbed the most water because their deeper roots had a competitive advantage across a larger soil water pool.

It should not be surprising that water uptake is correlated with plant growth, but this correlation does suggest root distributions are a primary determinant of water uptake and plant growth that  is not masked by factors such as water use efficiency, competitive exclusion, herbivory, dispersal limitation or fire sensitivity. The links between root distributions,

water uptake and plant growth have been demonstrated in plant monocultures and for whole communities, but demonstrating these effects for components of the plant community remains uncommon (Knapp et al., 2008; Holdo and Nippert, 2022; Lauenroth and Bradford, 2009; Sala et al., 2015). Yet, understanding how plant types use water is critical for predicting how plant types will respond to climate change or different species compositions (e.g., shrub encroachment or species invasions). Results suggest that predicting plant responses to climate change will require an understanding of how roots of different

species absorb water in different conditions.

In response to precipitation intensity treatments, shrubs and forbs grew better with larger precipitation events, which are predicted to occur more frequently in the future, while grasses did not. Larger precipitation events increased soil water availability by decreasing evaporation and increasing percolation. Shrub roots moved to depths that increased estimated water uptake 12 mm (from 93 mm yr$^{-1}$ to 105 mm yr$^{-1}$) and forb roots increased estimated water uptake 9 mm (from 79 mm

yr$^{-1}$ to 88 mm yr$^{-1}$) while the increase in grass uptake was only 5 mm (from 80 mm yr$^{-1}$ to 85 mm yr$^{-1}$). Thus, shrub roots increased their competitive advantage for water uptake, which correctly predicted a positive aboveground growth response by shrubs in high precipitation intensity plots (Fig. 3). Similarly, forbs moved their roots to depths that provided a competitive advantage for water relative to grass roots and this predicted the positive aboveground growth response of forbs to treatments (Fig. 4). Grass roots were also estimated to absorb more water (because more soil water was available), but

grass roots lost their competitive advantage to forbs, helping explain a lack of grass response to treatments. In short, a) rooting distributions affected the amount of water a plant could absorb, and b) the amount of water a plant could absorb





predicted plant abundance under ambient and manipulated precipitation conditions. These results are consistent with two previous studies that used the same techniques and found that root distributions could be used to predict plant landscape abundances (Kulmatiski and Beard, 2022; Kulmatiski et al., 2020a).

Shrub roots were estimated to provide a competitive advantage for water uptake, but our approach indicates several reasons that shrubs are not expected to competitively exclude forbs and grasses. First, shallow forb and grass roots have priority access to water as it enters the soil. This shallow rooting strategy by forbs and grasses did not provide the largest amount of water, but it did provide priority access to some water as it entered the soil (Kulmatiski and Beard 2022). Second, each rooting distribution demonstrated a competitive advantage at some soil depth (*i.e.*, the color-filled areas in Fig. 4). We

suggest these 'unique hydrologic' niches allow a plant to grow when rare (Ward et al., 2013; Kulmatiski et al., 2020a). Similarly, our calculations of water uptake assume that plants can absorb water at every soil depth in proportion to their relative abundance at that depth. In other words, we assume that plants can absorb water even when rare (in proportion to their relative abundance). The fact that we measured tracer uptake by all plants at all depths supports this assumption. Further, this assumption is consistent with studies demonstrating symmetric root competition (Cahill, Jr. and Casper, 2000;

Raynaud and Leadley, 2005). We suggest that the carbon costs associated with fully competitively excluding all other roots from all soil depths are too large, allowing different plants to access some water from most soil depths (Cabal et al., 2020; Cai et al., 2018). Further, the fact that water percolates deeply into the soil indicates that roots do not intercept all soil water. Consequently, there are likely to be precipitation pulses when water is not limiting during which all plants can absorb soil water.

It is important to understand the extent to which plant roots move in response to changing climate conditions (i.e., root plasticity; Guderle *et al.*, 2018; Liu *et al.*, 2018; Zhou *et al.*, 2019). Do some species move their roots in response to soil water availability while others maintain rigid rooting patterns (Berry and Kulmatiski, 2017)? Consistent with results from a similar study in a sub-tropical savanna, we observed that shrubs moved roots in response to treatments in ways that increased their water uptake (Berry and Kulmatiski, 2017; Kulmatiski and Beard, 2013). Results support the idea that the shrub

encroachment observed around the globe over the past 50 years is caused at least in part by the ability of woody plants to produce deep and flexible rooting patterns that can absorb more water as precipitation events become larger (Stevens et al.,

2017; Venter et al., 2018). We similarly found that forbs moved their roots in ways that increased water uptake in high precipitation intensity treatments. This provides a potential explanation for the increased forb growth observed in this and other experiments testing the effects of increased precipitation intensity (Jones et al., 2016). Predicting plant responses to climate change, therefore, is likely to require an understanding of how root distributions respond to changing soil resource availability.

It is interesting to note that plants absorbed more deep soil water in high precipitation intensity plots even though they moved their roots up the soil profile. This occurred because more deep soil water was available in high precipitation intensity plots, so even a small number of deep roots could absorb a large amount of water. That plants moved their roots up in the soil profile in response to increased deep soil water availability is a counterintuitive response that highlights the need for more direct measurements of root activity and water uptake to better understand root responses to climate change. This response is, however, consistent with the 'shallowest' rooting depth hypothesis (Schenk, 2008).

The shelters used in this experiment caused a small amount of warming (0.6° C) but had little effect on potential evapotranspiration due to lower windspeeds and higher relative humidity (Holdrege et al., 2021). A recent modeling study indicated that warming can have a larger effect than anticipated changes in precipitation event intensity (Holdrege et al., 2022) but see (Liu et al., 2020b; Volenec and Belovsky, 2018). Further research will be needed to better understand the combined effects of larger precipitation events and warmer temperatures (Giorgi et al., 2011; Jiao et al., 2021; Liu et al., 2018).

This study focused on the effects of vertical root distributions. There are many other factors that may improve understanding of plant responses to climate change. Grass stem conductance has been suggested to be larger than shrub stem conductance, but this adjustment alone would increase estimates of grass growth which would not improve our predictions (Holdo and Nippert, 2022). Aerodynamic properties of the canopy may improve predictions because shrubs and grasses tend to be taller and more exposed to air turbulence and low relative humidity that would decrease their water use efficiency and growth (Mazzacavallo and Kulmatiski, 2015). Whole plant water use efficiency may also have large effects on the eventual translation of water uptake into plant cover and biomass, but data on whole plant water use efficiency are lacking and mixed

(Hai et al., 2022; Toft et al., 1989; Golluscio and Oesterheld, 2007). These measurements will be needed for future efforts to convert our estimates of root water uptake to plant growth (Holdo and Nippert, 2022).

## 4.    CONCLUSIONS

Using a large-scale field experiment, we found that fewer, larger precipitation events increased forb and shrub growth while grass growth remained unchanged. Results, therefore, support the idea that precipitation intensification is likely to continue to contribute to shrub encroachment and forb growth in arid and semi-arid systems (Wilcox et al., 2018; Gherardi and Sala, 2015; Holdrege et al., 2022; Jones et al., 2016). We also described how active root distributions affect water uptake and demonstrate that these estimates of water uptake can explain plant abundance and coexistence on the landscape (Kulmatiski

and Beard, 2022; Kulmatiski et al., 2020a; Case et al., 2020). Further, we observed how plant roots responded to increased precipitation intensity and found that root responses helped predict plant community responses to climate change (Zhou et al., 2019; Liu et al., 2018). Taken together, results provide clear support for a central role of root distributions in determining plant growth and coexistence under current and anticipated climate conditions. Further use of the approach used here can be expected to improve predictions of arid and semi-arid ecosystem responses to climate change around the world.

**Code availability**

Upon acceptance, code used in this manuscript will be given a DOI and made available at the USU Digital Commons.

**Data availability**

Upon acceptance, data used in this manuscript will be given a DOI and made available at the USU Digital Commons.


**Executable research compendium**

Not applicable

**Sample availability**

Not applicable

**Video supplement**



Not applicable

**Team list**

Not applicable

**Author contribution**

AK conceived and performed the experiment, analyzed data, and wrote the paper. MCH performed the experiment and

analyses and edited the paper. CC performed the experiment and analyzed data. KHB conceived the experiment and edited

the paper.

**Competing interest**

The authors declare that they have no conflict of interest.


**Disclaimer**

Not applicable

**Acknowledgements**

This research was supported by the Utah Agricultural Experiment Station and approved as journal paper #9631, Utah State

University, and the Utah State University Ecology Center. We thank A. Crow, B. Gunderson, S. Hall, L. Henzler, N.

Macriss, S. Sprouse, C. Walton, T. Wiese, and W. Wilson for field assistance; S. Durham for statistical advice; and Brad

Hunt of the Hardware Ranch Wildlife Management Area.

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

**Table 1** AIC table for models of tracer uptake by depth for a factorial combination of season (early or late) and precipitation intensity (low intensity or high intensity). For the 'All together' model, measurements from the three measured plant functional groups (forbs, grasses and shrubs) were not distinguished. For the 'All separate' model, measurements were associated with forbs or grasses or shrubs.





| Model | logLik | AIC | ΔlogLik | ΔAIC | df |
|---|---|---|---|---|---|
| Early/ low | | | | | |
| All together | 369.7 | -731.4 | 0.0 | 0.0 | 4 |
| All separate | 376.4 | -732.8 | 6.7 | 1.4 | 10 |
| Early/ high | | | | | |
| All together* | 386.3 | -764.6 | 3.6 | 4.7 | 4 |
| All separate | 389.9 | -759.9 | 0.0 | 0.0 | 10 |
| Late / low | | | | | |
| All together | 380.2 | -752.4 | 0 | 0 | 4 |
| All separate* | 388.6 | -757.3 | 4.9 | 7.6 | 10 |
| Late / high | | | | | |
| All together | 401.6 | -795.1 | 0.0 | 0.0 | 4 |
| All separate* | 420.1 | -820.3 | 18.5 | 25.2 | 10 |

Abbreviations: logLik, log likelihood; AIC, Akaike's information criterion; df, degrees of freedom.

*Indicates top model based on ΔAIC < 2 criteria.

**Figure legends**

**Figure 1**. **Photographs of precipitation manipulation shelters on site**. Eleven precipitation manipulation shelters (8 m x 8 m) with plastic roofs collected and redeposited precipitation through as sprinkler system (a) and shoveling (b) as fewer, larger events. Tracer injections (a) were performed in plots to describe active root distributions of forbs, grasses and shrubs. Treatments were applied between January 2016 and September 2020 and vegetation growth measurements continued through 2021, one year after treatment removal.


**Figure 2. Soil volumetric moisture content in low precipitation intensity and high precipitation intensity plots**. Data recorded every 20 minutes at six depths in one low precipitation intensity plot and one high precipitation intensity plot.

**Figure 3. The proportion of tracer uptake by soil depth for forbs, grasses and shrubs in either low or high intensity** 590 **precipitation manipulation plots during early (May) and late (July) season samplings**. Each point represents the mean uptake value from four different plots. Error represents variation among four experimental plots. These values used to inform the rooting profiles in a soil water flow model.





**Figure 4. Water uptake by forbs, grasses, and shrubs by soil depth in low precipitation intensity treatment plots (a)**
**and high precipitation intensity treatment plots (b).** Data represents the average annual amount of water each rooting

distribution was estimated to absorb if each rooting distribution was attached to a similar plant canopy (i.e., same leaf area

and stomatal conductance). Color-filled areas highlight water uptake for which the indicated plant type has a competitive

advantage. For example, the black area between 40 cm and 100 cm in panel a indicates that shrubs have a competitive

advantage for water uptake at these depths in the low precipitation plots. Grey areas indicate shared soil water. The sum of

water uptake across the profile in low precipitation intensity conditions was 79 mm, 80 mm and 93 mm, for forbs, grasses

and shrubs, respectively. The sum of water uptake in high precipitation intensity plots was 88 mm, 85 mm, and 105 mm, for

forbs, grasses and shrubs, respectively.

**Figure 5. Water uptake estimated for forbs, grasses and shrubs as a function of root distributions (Fig. 4) and**
**measured leaf area and stomatal conductance for each plant type in plots with a) low precipitation intensity and b)**
**high precipitation intensity**. The sum of water uptake across the profile for forbs, grasses and shrubs was 42 mm, 77 mm

and 140 mm of water per year in low precipitation intensity conditions and 46 mm, 82 mm and 158 mm per year in high

precipitation intensity conditions, respectively.


**Figure 6. Shrub stem radius (+/- 1 SE) over time in low and high precipitation intensity plots over five years.** Stem

diameter recorded hourly on three shrubs in six low precipitation intensity plots and five high precipitation intensity plots.

All plots received the same total annual precipitation. Precipitation treatments began January 2016 and continued to

September 2020 as indicated by vertical dashed lines indicate. There was more support for models that separated treatments

than for a model that combined treatments, both during and after treatment applications ($\Delta$ AIC = 1082 and 2047,

respectively).



**Figure 7**. **Forb (a), grass (b) and shrub (c) ground cover response to low and high intensity precipitation treatments over time**. Dashed vertical lines indicate the beginning and end of treatment application. Error derived from variation among
six low precipitation intensity plots and five high precipitation intensity plots. All plots received the same total annual precipitation.


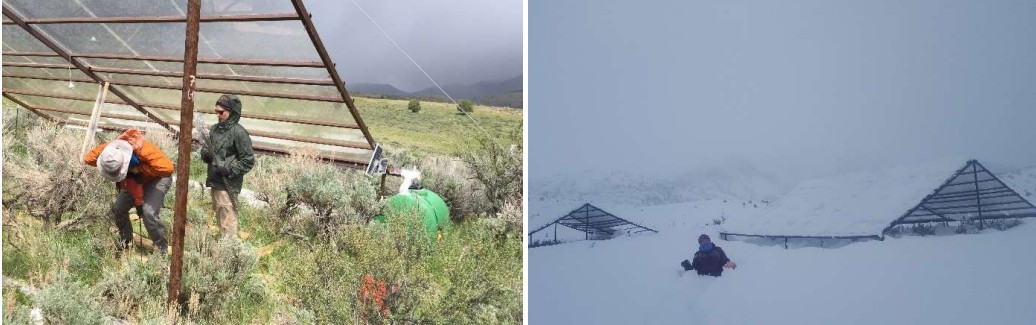

**Fig. 1**. Eleven precipitation manipulation shelters (8 m x 8 m) with plastic roofs collected and redeposited precipitation through as sprinkler system (a) and shoveling (b) as fewer, larger events. Tracer injections (a) were performed in plots to describe active root distributions of forbs, grasses and shrubs. Treatments were applied between spring 2016 and fall 2020
and vegetation growth measurements continued through 2021, one year after treatment removal.



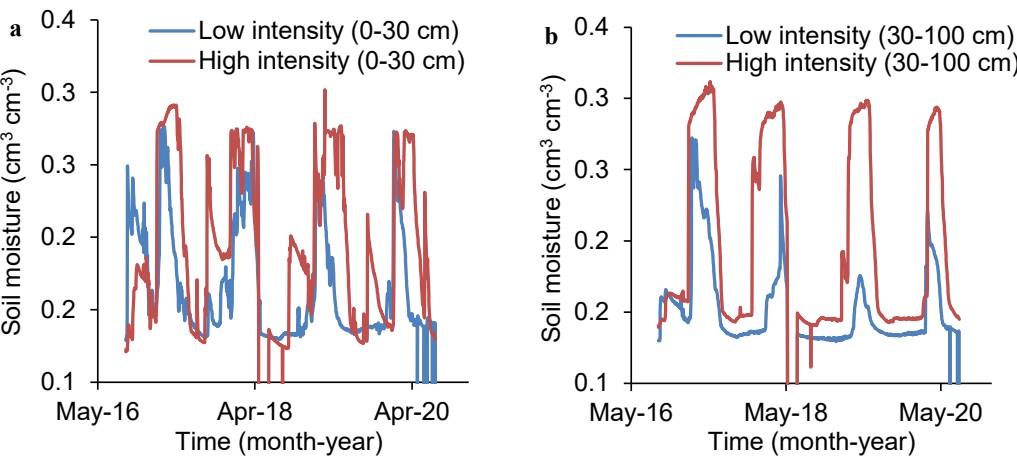

**Fig. 2.** Soil volumetric moisture content in low precipitation intensity and high precipitation intensity plots. Data recorded

every 20 minutes at six depths in one low precipitation intensity plot and one high precipitation intensity plot.

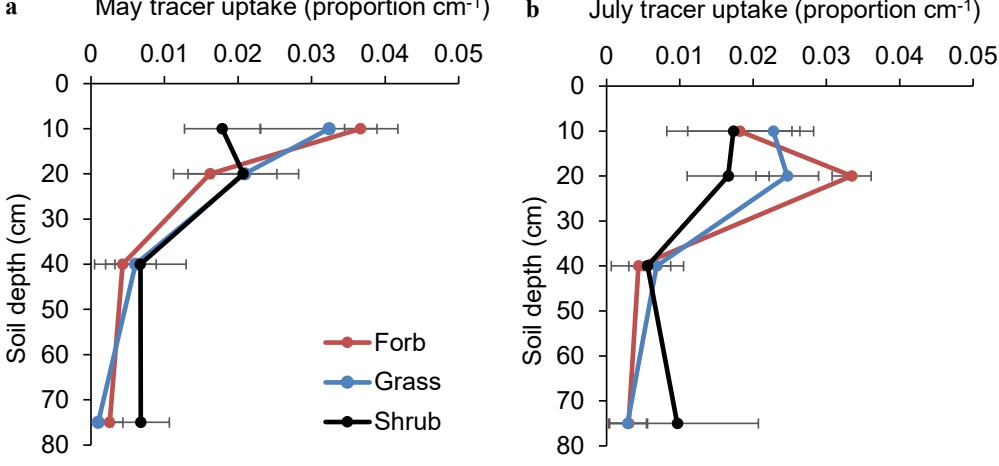

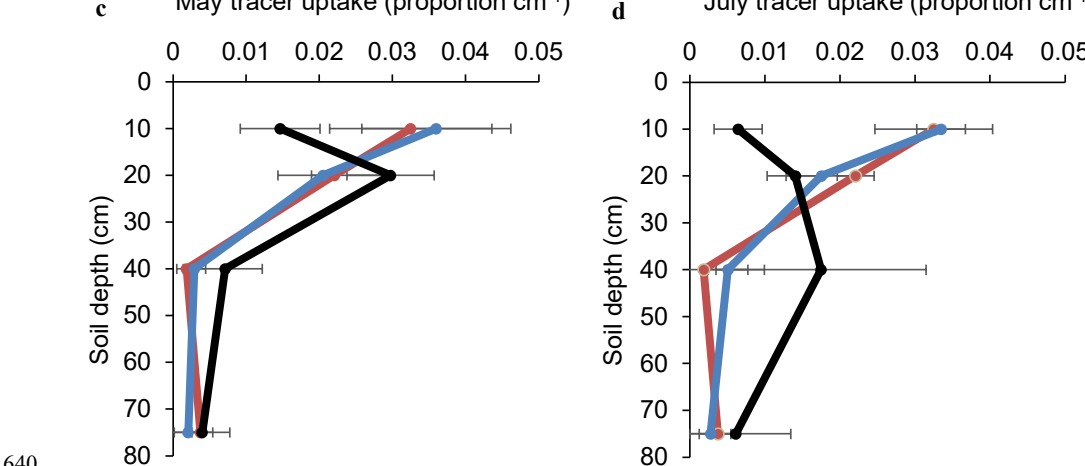


**Fig. 3.** The proportion of tracer uptake by soil depth for forbs, grasses and shrubs in either low (a,b) or

high (c,d) intensity precipitation manipulation plots during early (May) and late (July) season

samplings. Each point represents the mean uptake value from four different plots. Error represents

variation among four experimental plots. These values used to inform the rooting profiles in a soil water

flow model.



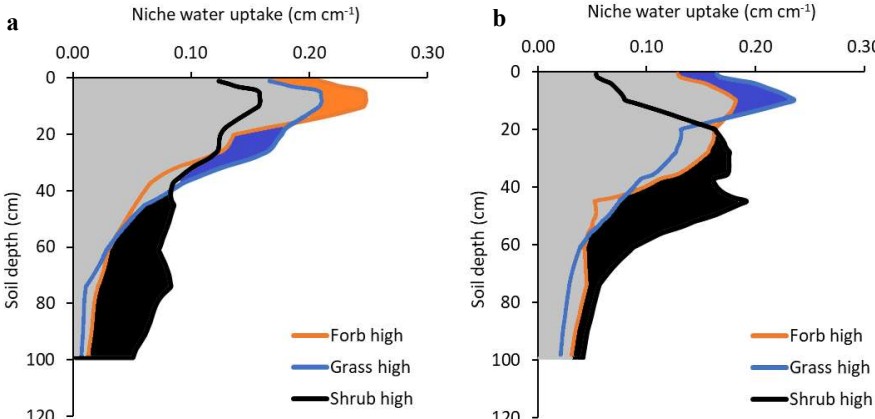

**Fig. 4.** Water uptake by forbs, grasses, and shrubs by soil depth in low precipitation intensity treatment

plots (a) and high precipitation intensity treatment plots (b). Data represents the average annual amount

of water each rooting distribution was estimated to absorb if each rooting distribution was attached to a

similar plant canopy (i.e., same leaf area and stomatal conductance). Color-filled areas highlight water

uptake for which the indicated plant type has a competitive advantage. For example, the black area

between 40 cm and 100 cm in panel a indicates that shrubs have a competitive advantage for water

uptake at these depths in the low precipitation plots. Grey areas indicate shared soil water. The sum of

water uptake across the profile in low precipitation intensity conditions was 79 mm, 80 mm and 93 mm,

for forbs, grasses and shrubs, respectively. The sum of water uptake in high precipitation intensity plots

was 88 mm, 85 mm, and 105 mm, for forbs, grasses and shrubs, respectively.





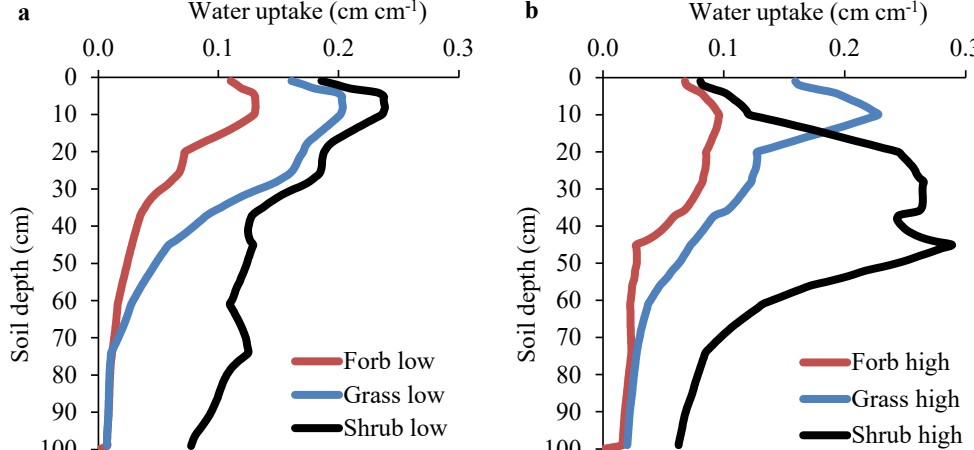

**Fig. 5.** Water uptake estimated for forbs, grasses and shrubs as a function of root distributions (Fig. 4) and measured leaf area and stomatal conductance for each plant type in plots with a) low precipitation intensity and b) high precipitation intensity. The sum of water uptake across the profile for forbs, grasses and shrubs was 42 mm, 77 mm and 140 mm of water per year in low precipitation intensity conditions and 46 mm, 82 mm and 158 mm per year in high precipitation intensity conditions, respectively.

670



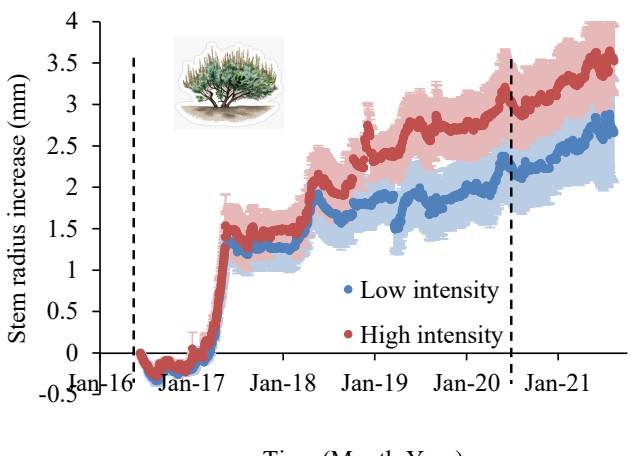

**Fig. 6.** Shrub stem radius (+/- 1 SE) over time in low and high precipitation intensity plots over five years. Stem diameter recorded hourly on three shrubs in six low precipitation intensity plots and five high precipitation intensity plots. All plots received the same total annual precipitation. Precipitation 675 treatments began May 2016 and continued to September 2020 as indicated by vertical dashed lines indicate. There was more support for models that separated treatments than for a model that combined treatments, both during and after treatment applications (Δ AIC = 1082 and 2047, respectively).

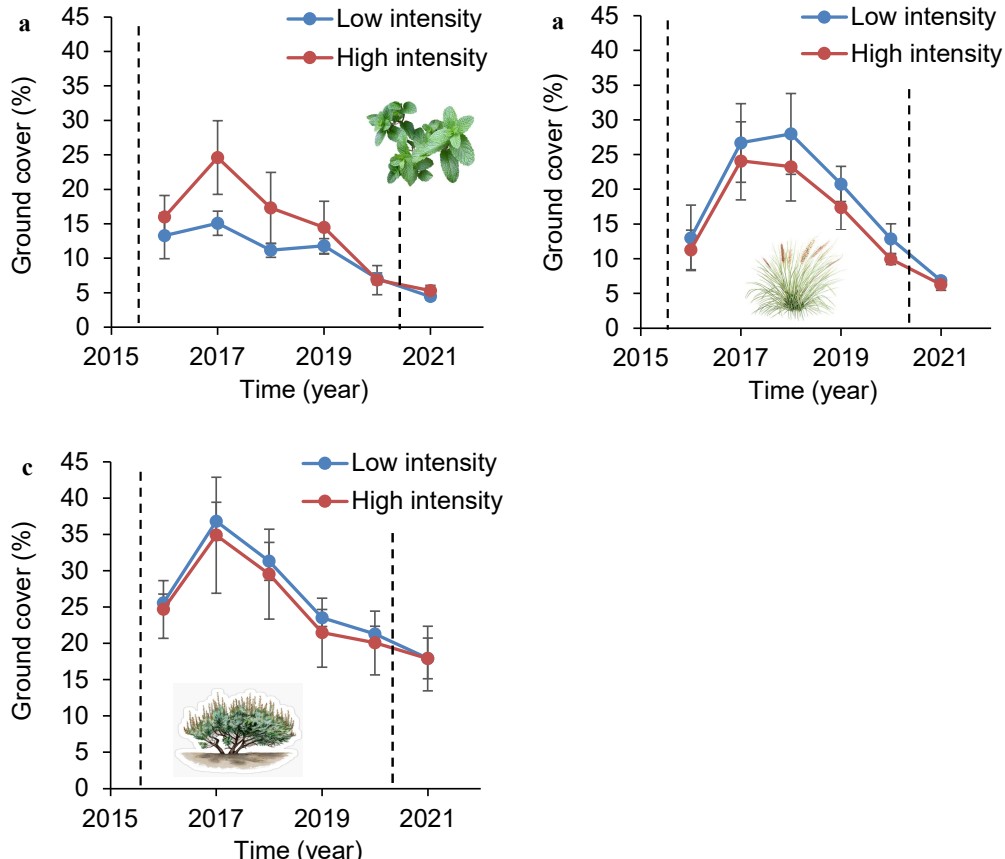

680

**Fig. 7.** Forb (a), grass (b) and shrub (c) ground cover response to low and high intensity precipitation

treatments over time. Dashed vertical lines indicate the beginning and end of treatment application.

Error derived from variation among six low precipitation intensity plots and five high precipitation

685    intensity plots. All plots received the same total annual precipitation.