# Peer review of "Root distributions predict shrub-steppe responses to precipitation intensity"

_Biogeosciences, 2023_

## Author Response (AR1)

**Public justification (visible to the public if the article is accepted and published)**:
The submitted paper is an interesting and relevant paper, as was also commented on by RC1 and RC2. The comments by RC1 and RC2 are relevant, and well addressed by the authors.

The author comments for RC1 were thorough, though I also missed a few things.
RC1 mentions that a change in root phenology could also result in the signal of the distribution of water uptake moving, but they come with a different C-economy. They ask to include this in the discussion, but I did not read a reply to this point in the author comments?

Response: I think there was some confusion during the review process. We have made the changes noted in our responses to reviewers, but we have not, until now been able to upload the revised version, so I think that the changes we have made were not evident to the editors and reviewers.

Broadly, to address the editor's comments, we have removed the LAI weighted data from the paper and we have elaborated on the phenology effects and carbon costs of different mechanisms of root water uptake. We think that you will find we have fully addressed reviewer and editor comments.

Response: We have added the following paragraph to address this comment:

"Tracer uptake indicated that root distributions changed over time and in response to increased precipitation intensity, but it does not indicate how plants cause these changes. Plants can grow new roots at specific depths, senesce roots at some depths more than others, or change aquaporin abundance or activity (Dybzinski et al., 2019; Zarebanadkouki et al., 2019). Similarly, a plant functional group could demonstrate deeper roots later in the season because all plants grow new deep roots or because some shallow-rooted species senesce early in the season. Our measurements did not distinguish these different mechanisms, but they may have important consequences for species carbon budgets or long-term growth and coexistence."

We have also elaborated on the carbon costs as follows in the Discussion: 'For example, a shrub would have to produce a large amount of very shallow roots to compete with dense grass root mats. This would come at a large direct carbon cost. At the same time, this direct carbon cost would come at a large opportunity cost of producing deep roots that provide more water that would allow a shrub to compete with other shrubs.'

L109-110, the new wording did not add much clarity, could you rephrase again?

We have added an appendix that shows the distributions of precipitation events and the coefficient of variation of precipitation event sizes (a standard approach for summarizing precipitation patterns).

To clarify this sentence, we have reworded this sentence as follows: 'The distributions of precipitation events and the coefficient of variation in precipitation events are described in the Appendix.'

L119-122, in L109, it is said that 4mm is linked to 3 degree warming, and not 5 degree warming?

Yes, this is correct -1C = 1mm (low), control = control (low), 1C = 2 mm (low), 2C = 3mm (low), 3C = 4 mm (high), 5C = 8mm (high), 10C = 18 mm (high).

To clarify, we have added the following "Because storms often resulted in more than one minimum event per day, the mean daily event sizes associated with treatments were 4.8, 5.3, 6.2, 7.2, 8.4, 10.8 and 19.4 mm for the 1, control, 2, 3, 4, 8 and 18 mm treatments (Holdrege et al. 2020)."

L167-169, add 'by depth' at the last part of the sentence as well?

Response: We have added 'by depth' as suggested.

the comment on L298-300 is also addressed by RC2. In the author comments for RC2, it is clear that the authors changed the manuscript quite a bit regarding the results weighted or not weighted by LAI. This is a major revision, that will will likely improve the manuscript, but is currently not completely clear nor visible.

Response: We have removed the LAI weighted measurements from the manuscript.

We find the weighted values important because they produce estimates of water flow in the system that can be verified with soil water data. They also allow us to compare our best estimate of water uptake to plant abundance on the landscape. This demonstrates whether our estimates are higher or lower than observed cover. If our estimates of water uptake are higher than observed cover of a functional group, it suggests that water use efficiency or herbivory suppresses growth of that functional group. We have removed these values since they cause confusion for the readers.

The first author response to RC2 did not include all points, but the second author response was thorough and relevant.

Additional private note (visible to authors and reviewers only):
I think the manuscript is interesting and relevant for the scientific community. The major revision regarding the results weighted or not weighted by LAI are described, but it currently is not completely clear how this will work out in the revised manuscript. I therefor accept the manuscript with major revisions.

Respons: Again, we have removed the LAI weighted values to avoid any confusion.